# Controlling Microparticle Morphology in Melt-Jet Printing of Active Pharmaceutical Ingredients through Surface Phenomena

**DOI:** 10.3390/pharmaceutics15082026

**Published:** 2023-07-26

**Authors:** Shachar Bornstein, Almog Uziel, Dan Y. Lewitus

**Affiliations:** Department of Polymer Materials Engineering, Shenkar College of Engineering, Design and Art, Ramat Gan 5252626, Israel; shacharb550@gmail.com (S.B.); almoguziel@gmail.com (A.U.)

**Keywords:** melt-jet printing, active pharmaceutical ingredients (APIs), particle engineering, microparticles, sphericity, surface properties

## Abstract

Achieving homogeneity and reproducibility in the size, shape, and morphology of active pharmaceutical ingredient (API) particles is crucial for their successful manufacturing and performance. Herein, we describe a new method for API particle engineering using melt-jet printing technology as an alternative to the current solvent-based particle engineering methods. Paracetamol, a widely used API, was melted and jetted as droplets onto various surfaces to solidify and form microparticles. The influence of different surfaces (glass, aluminum, polytetrafluoroethylene, and polyethylene) on particle shape was investigated, revealing a correlation between substrate properties (heat conduction, surface energy, and roughness) and particle sphericity. Higher thermal conductivity, surface roughness, and decreased surface energy contributed to larger contact angles and increased sphericity, reaching a near-perfect micro-spherical shape on an aluminum substrate. The integrity and polymorphic form of the printed particles were confirmed through differential scanning calorimetry and X-ray diffraction. Additionally, high-performance liquid chromatography analysis revealed minimal degradation products. The applicability of the printing process to other APIs was demonstrated by printing carbamazepine and indomethacin on aluminum surfaces, resulting in spherical microparticles. This study emphasizes the potential of melt-jet printing as a promising approach for the precise engineering of pharmaceutical particles, enabling effective control over their physiochemical properties.

## 1. Introduction

The production of solid powders from synthesized APIs involves a series of steps aimed at obtaining high-quality particles with desired properties. One crucial aspect is the crystallization of the API in reactors to obtain a crystal slurry through cooling, evaporation, seeding, or antisolvent addition. This is followed by solid–liquid separation, washing, and drying field [1,2]. Each solid form of the API has unique properties that affect its solubility, bioavailability, hygroscopicity, melting point, stability, flowability, compressibility, and other performance characteristics [3]. Controlling these processes and understanding their impact on the resulting drug properties is crucial in the pharmaceutical industry as API particles’ physicochemical properties and solid-state morphology directly affect their performance [2,3].

Various methods are utilized in the pharmaceutical industry to produce particulate materials, and new techniques are constantly being developed [4]. Most methods aim to manufacture drug delivery systems by combining the API with an excipient carrier [5]. Among these methods, spray drying is the most prevalent technique, particularly for generating inhalable particles using specialized excipients and isolating thermally labile products [6]. However, spray drying is rarely used to produce pure API particles due to the inherent challenges of partial dissolution, which can lead to uncontrolled recrystallization and chemical instability.

An alternative technique introduced recently is electro-spraying, which enables the production of pharmaceutical particles. This method involves the application of a high voltage (several kilovolts) to a microcapillary nozzle in a spraying system [5]. When used for producing drug particles or drugs combined with an excipient, a solution is first prepared in an appropriate organic solvent. Then, it is sprayed, frequently with the addition of an electrolyte, into a receiving solution containing additional stabilizers to prevent particle aggregating after formation; thus, the generated API particles are not pristine. Moreover, it has limitations in achieving precise control over particle size, size distribution, and morphology [5,7].

Another method employed for creating drug particles or delivery systems within the micrometer size range is spray cooling, also called spray congealing [8,9]. In this technique, the molten material is atomized into a cold, inert environment (usually nitrogen), causing the droplets to solidify and form spherical, free-flowing particles suitable for direct use in tableting or capsule filling. This eliminates the need for additional downstream processes such as secondary drying, milling, or granulation. However, spray congealing has certain drawbacks, including relatively high process temperatures, a narrow range of particle sizes, and high maintenance costs [9]. Moreover, spray congealing is rarely used to process the API alone and is usually accompanied by a molten excipient. Furthermore, the downscaling of the process, making it suitable for the spraying and generation of small amounts of API, or reproducing the process in a laboratory environment, poses challenges.

Printing technologies have emerged as a promising approach in pharmaceutical manufacturing, particularly for drug delivery systems. While there are numerous examples of digital manufacturing of dosage forms that include an API with an excipient [10,11], printing pure API particles is still rare, especially when aiming for the precise engineering of particle properties. Among these rare occasions of digital printing of API, inkjet printing (IJ) has demonstrated potential in producing drug-only systems by preparing ink by dissolving the API in an appropriate solvent. IJ is a non-contact approach that allows for the precise processing of liquid droplets on a substrate without contact with two-dimensional and three-dimensional structures [12,13]. Various dosage forms have been created using 2D and 3D IJ, generating API films for several applications, including localized or targeted action, improved efficacy, safety, reduced toxicity, and, consequently, improved patient compliance [14,15].

Recently, we have shown that melt droplet deposition (melt-jet printing) using a specialized valve allows for the fabrication of solid spherical particles from melts. This technique eliminates the need for organic solvents and surfactants, offering several advantages in the production of drug particles [16,17,18]. Melt-jet printing involves depositing a controlled amount of molten material onto a cold non-wetting surface, resulting in immediate solidification. Using this technique, polycaprolactone-based microspheres with varying amounts of ibuprofen as a model drug were produced. In vitro release studies revealed that one can control the crystal characteristics of both the excipient polycaprolactone and ibuprofen API, through the precise control of the printing parameters, with the substrate temperature being the most significant, resulting in tunable drug release rates over extended periods. The melt-jet printing approach offers several advantages: controlled particle size via nozzle diameter, controlled crystallization through the cooling rate, minimal material loss, and solvent-free processing, which are inherent to the melt-based process, and reproducibility. Additionally, we discovered that the interaction between the jetted-molten droplet and surface influences the resulting particle shape (sphericity). The substrate’s roughness, surface energy, along with the surface tension of the melt have been found to be crucial factors affecting the shape of the solidified particle [16,19,20,21,22].

In this study, we evaluated the capacity to apply the melt-jet printing technique to produce pristine paracetamol raw material (powder) directly from crude while controlling several API properties in a single step, including shape, size, and polymorphism, thus deeming them suitable for use in the preparation of dosage forms. This was achieved by depositing paracetamol molten droplets onto different surfaces while elucidating whether a relationship between melt printing parameters, including substrate properties, would affect the API microparticle shape and morphology. This was achieved through the melt deposition of the molten API onto various surfaces with a wide range of physio-chemico-thermo properties and the investigation of the potential correlations between these surface characteristics and the formed solid particle’s shape and morphology. Factors such as surface energy, roughness, chemical composition, and thermal conductivity were considered. Thermal and chemical characterization methods were employed to evaluate whether the printing process itself resulted in changes to the integrity of the paracetamol, including HPLC, DSC, and XRD. Furthermore, the method’s applicability to other APIs was demonstrated by printing both carbamazepine and indomethacin on chosen surfaces.

## 2. Results and Discussion

### 2.1. Melt-Jet Printing of Paracetamol Microparticles on Different Surfaces

Paracetamol microparticles were produced using a melt-jet printing technique, wherein the molten droplets were deposited onto various surfaces. Our previous research has indicated that the interaction between the surface and the molten droplets plays a crucial role in achieving spherical particle formation [16]. Hence, we selected four distinct surfaces for this study: two polymeric surfaces, polytetrafluoroethylene (PTFE) and polyethylene (PE); one ceramic surface, glass; and one metallic surface, aluminum. These surfaces vary in their surface energy (tension), surface roughness, thermal conductivity, and heat capacity. The objective was to investigate whether the substrate material would have an effect on the printing outcome in terms of the particle shape and morphology to allow better control over these parameters. Figure 1 shows representative scanning electron microscope (SEM) images that depict the paracetamol printed particle shape as a function of the varying printing substrate material. By keeping all other printing parameters the same, we observed that altering the substrate alone resulted in distinct differences in the particle shapes, particularly in terms of their sphericity [23]. Notably, the particles exhibited a flattened bottom on the PTFE, PE, and glass surfaces, while the particles printed on the aluminum surface exhibited a spherical shape. These findings suggest that the molten droplets reach the surfaces while they are still in a molten state (in contrast to spray congealing, where they solidify in the air). As result, the molten droplets are allowed to effectively wet the surface prior to solidifying into particles [16]. A notable difference observed among the particles was the measurement of their apparent contact angles obtained from the SEM images. These were determined to be 88.5 ± 4.1, 104.5 ± 1.5, 131.4 ± 15, and 180 ± 0 degrees for PTFE, glass, PE, and aluminum surfaces, respectively.

### 2.2. Influence of Surface Properties on Paracetamol Microparticle Shape in Melt-Jet Printing

To further elucidate the relationship between melt-jet printing on different surfaces and the resulting microparticle shapes, an inquiry was conducted to identify potential correlations between surface structure and properties and the resulting form of the printed microparticles. Surface wetting, a phenomenon influenced by various surface factors such as surface energy, roughness, and chemical composition, plays a crucial role in determining the contact angle and, consequently, the wettability of the material surface [24]. The relationship between surface roughness (Ra) and wettability, as defined by Wenzel, suggests that surface roughness can improve wettability when combined with surface chemistry. For instance, a chemically hydrophobic surface becomes even more hydrophobic with the addition of surface roughness. High surface energy (HSE) results in strong molecular attractions and a lower contact angle between the surface and the microparticles. In contrast, low surface energy (LSE) results in weak molecular attraction, which makes bonding more challenging [25]. Thus, we attempted to correlate these aforementioned surface properties to the resulting particle shapes. The roughness of the surfaces was measured using atomic force microscopy (AFM). Representative AFM images of the four different surfaces (PTFE, PE, aluminum, and glass) are shown in Appendix A. The mean surface roughness values were found to be highest for the PE and lowest for the glass. The surface energy values of the substrates were found in the literature and were highest for aluminum and lowest for PTFE. Thus, no correlation between these two parameters and the particle’s shape and sphericity was found (see table in Figure 2a). Consequently, an additional material property plays a role in determining the droplets interaction with the surface. Zitzenbacher et al. [26] explored the contact angles of molten polymers on different coating materials and observed a decrease in the contact angle between the melt and the surface with an increase in the surface energy of the coating. It was also noted that the contact angle of polymer melts on polished steel decreased as the surface temperature increased. These findings suggest that as the thermal conductivity of the substrate increases, leading to a higher cooling rate, the temperature of the melt decreases, resulting in larger contact angles and increased sphericity. In addition, Zhou et al. [21] found that increasing an aluminum substrate’s temperature enhances the interaction between a polypropylene droplet and the substrate, resulting in an increased contact area between the PP droplet and the substrate, subsequently leading to a smaller contact angle. Materials with high thermal conductivity (K) can effectively transfer and dissipate heat [27], leading to faster cooling and the maintenance of spherical microparticles. Furthermore, Zitzenbacher et al. [20] measured the contact angle of polymer melts on polished steel, finding that the contact angle decreases with increasing surface temperature due to the decrease in surface tension of the liquid polymer melt. Thus, in addition to the surface roughness and surface energy, the thermal conductivity of the four substrates was collected from the literature [28,29,30,31] to allow us to assess the parameters influencing the wetting phenomena on paracetamol particle formation. The properties of the surfaces and their corresponding apparent contact angles are detailed in Figure 2.

These findings suggest that multiple surface factors come into play when a molten API droplet interacts with a surface. The contact angle and microparticle shape are influenced not only by surface roughness and surface tension, but by the substrate’s thermal conductivity as well. As thermal conductivity increases, leading to a higher cooling rate, the melt temperature decreases, resulting in larger contact angles, in turn leading to an increase in sphericity. These observations demonstrate that greater thermal conductivity and surface roughness contribute to larger microparticle contact angles, yielding rounder-shaped particles. Conversely, higher substrate surface energy leads to smaller contact angles and flatter microparticles. This explains the variations in particle shape observed when printing on surfaces with different properties. Furthermore, these findings can be visualized by plotting the apparent contact angle against an expression of the thermal conductivity, surface roughness, and surface energy, as depicted in Figure 2b.

During the melt-jet printing process, a specialized inkjet-like device ejects a small portion of a molten API onto a surface. The device operates by applying pressure to the molten material, causing it to be expelled as a droplet. Initially, an ejected droplet with a following tail is formed [32]. Then, as the ejected droplet interacts with the surface, the tail merges with the main drop, leading to the spontaneous formation of a spherical shape on the surface. Figure 3 shows a sequence of images captured using a high-speed camera emphasizing the spontaneous formation of paracetamol microparticle upon the interaction of the molten jet with the aluminum surface.

### 2.3. Characterization of Melt-Jet-Printed Paracetamol Microparticles

Spherical particles were successfully obtained exclusively on aluminum surfaces, leading us to further characterize them. Figure 1j–l shows the SEM images of paracetamol microparticles printed on an aluminum surface, revealing a smooth surface and a spherical shape with an average particle size of 207 ± 35 μm.

To analyze the polymorphism of the paracetamol and ensure that the microparticle production process did not induce polymorphic form changes, differential scanning calorimetry (DSC) measurements were conducted. DSC is a valuable technique for characterizing polymorphism in pharmaceutical materials, specifically for determining form I from form II in paracetamol [33]. Figure 4a,b present the DSC results obtained for the neat paracetamol powder and the printed paracetamol particles. Before processing, the neat paracetamol exhibited an endothermic peak at 171.2 °C with a melting enthalpy of 162.8 J/g. In comparison, the printed microparticles displayed an endothermic peak at 170.7 °C with a melting enthalpy of 169.8 J/g, with no apparent shoulder at 140 °C or skewing of the curve to lower temperatures, indicating that they retained the monoclinic form I of paracetamol without significant changes from the neat powder [33]. The polymorphic form of an API plays a critical role in determining its physical characteristics and subsequent therapeutic efficacy. The introduction of a new or unknown crystal form can have significant implications for product performance, occasionally leading to detrimental clinical outcomes [33]. Paracetamol exists in three polymorphic forms: two stable forms, monoclinic (form I) and orthorhombic (form II), along with one unstable form (form III). Understanding and controlling the polymorphic forms can influence factors such as dissolution rate, ease of consumption, and API release rate [34]. To further attest to the effect of the printing process on paracetamol’s morphology, XRD analyses were performed. XRD results, presented in Figure 4c, reveal that the printed particles exhibit a pattern resembling those of the neat powder. The reflexes in the reference appear to be less broad compared to the printed material, which may be attributed to a smaller crystallite size in the printed material resulting from the rapid cooling during the printing process [35]. Additionally, relative reflex intensities differ between the two materials (e.g., near 20° 2θ) potentially due to preferred orientation and large crystals in the neat powder [36], while for the printed spherical particles, the presence of a preferred orientation is unlikely.

Paracetamol is a synthetic non-opiate derivative of the toxic 4-aminophenol, with 4-aminophenol being its primary hydrolytic product [37]. Various methods have been reported for assessing paracetamol’s purity and detecting its main degradation products, including high-performance liquid chromatography (HPLC) [38]. Figure 4d displays HPLC chromatograms of the paracetamol powder and microparticles (shifted for clarity). In both chromatograms, a prominent paracetamol peak appears at 5.5 min. In the printed microparticles, two additional peaks are visible at 28.8 and 33.1 min. However, given that their measured quantities are less than 0.05% of the paracetamol peak, as per the European Pharmacopoeia [39], they are considered negligible, suggesting that no significant degradation products were generated during the particle production. It is possible that thermally sensitive APIs may exhibit more significant degradation products after the printing process. To evaluate their stability, printed particles were stored under various temperature and humidity conditions for 4 weeks. DSC curves of analyses performed at different storage conditions indicated that no major changes in the polymorphic structure of paracetamol were observed (Appendix A). These findings are further corroborated by the XRD analyses (Appendix A). Moreover, all the HPLC chromatograms (Appendix A) showed a prominent paracetamol peak with no significant degradation products. Lastly, SEM images of the particles after 4 weeks are shown in Appendix A. As observed, no major changes were seen, demonstrating that the particles retained their original morphology and structure.

To further demonstrate the method’s applicability to other APIs, carbamazepine and indomethacin were also printed on aluminum surfaces. These materials exhibit different melting temperatures and were processed at a reservoir temperature of 165 °C for the indomethacin and 200 °C for the carbamazepine, compared to the paracetamol’s processing temperature of 176 °C. Figure 5 shows the SEM images of near-perfect spherical microparticles of paracetamol, carbamazepine, and indomethacin printed on an aluminum surface, indicating that regardless of the API used, all particles exhibited consistent spherical shapes. This finding highlights the robustness and versatility of the melt-jet printing method in achieving uniform particle shapes among various APIs.

## 3. Materials and Methods

### 3.1. Materials

Paracetamol, carbamazepine, and indomethacin were kindly provided by Merck KGaA, Darmstadt, Germany. N,N-Dimethylformamide, ethanol (HPLC grade), methanol (HPLC grade), acetone, and dimethyl sulfoxide (DMSO) (HPLC grade) were purchased from Bio-Lab Ltd., Jerusalem, Israel.

### 3.2. Preparation of Microparticles

API particles were prepared in a similar manner to polymeric micro-particles [16,18], with some modifications. API was inserted into a heated reservoir of a pneumatic jetting valve (P-Jet CT, Liquidyn^®^ (Nordson EFD) Oberhaching, Germany). The following settings were used for paracetamol, indomethacin, and carbamazepine, respectively: reservoir temperature: 176 °C, 165 °C, and 200 °C; die temperature: 174 °C, 163 °C, and 198 °C; valve pressure: 0.85 bar, 1.8–1.9 bar, and 1–1.2 bar. The spring (tappet) tightness was 0.6–0.9 turns, and the distance between the nozzle and the surface was set at 30 cm. The temperature and pressure were controlled using a Liquidyn V100 controller. To generate micro-particles, droplets of the molten API were jetted through a 100 µm needle flat die nozzle (Liquidyn^®^) onto an aluminum, PTFE, PE, or glass surface and subsequently cooled at room temperature.

### 3.3. Microparticle Characterization

Microparticle size, shape, surface morphology, and apparent contact angle were evaluated using scanning electron microscopy (SEM). The microparticles were coated with an Au-Pd mixture using a sputter coater unit (SC7620, Quorum, East Sussex, UK). Microparticle size and surface morphology were assessed using a JEOL JSM-IT200 SEM (JEOL, Tokyo, Japan) operated at an acceleration voltage of 5 kV. The microparticle size was measured using the SEM software (JSM-IT200 Operation software) roller. The apparent contact angle between the microparticles and the surface was determined using the SEM’s angle measurement function. The measurements were conducted on 10 randomly selected particles, and the average and standard deviations were reported.

### 3.4. Thermal Analysis

Microparticles and neat powder of API (5–10 mg) were analyzed via differential scanning calorimetry (DSC) measurements (Q200, TA Instrument, New Castle, DE, USA). All experiments were performed from 25 to 230 °C at a scanning rate of 10 °C/min, with one heating run using closed crucibles.

### 3.5. HPLC Analysis of Paracetamol

HPLC analysis was carried out using a UHPLC system (Thermo Scientific, Bremen, Germany) fitted with a C18 core–shell column (2.6 μm, 150 mm × 2.1 mm i.d.) and a SecurityGuard Ultra column (2 mm × 2.1 mm i.d) (Phenomenex, Torrance, CA, USA). Chromatographic separation was achieved using a previously described method [39] with some adjustments. Separation was done with a solvent mixture of phosphate buffer (prepared by dissolving 1.7 g of potassium dihydrogen phosphate and 1.8 g of dipotassium hydrogen phosphate in HPLC-grade water and diluting to 1000 mL with water) (solvent A) and methanol (solvent B). The gradient profile started at 5% B for 1 min and was then raised to 10% over 9 min, held at 10% for 10 min, then raised to 34% over 20 min, and held at 34% for 10 min. At 50 min, the column was returned to 5% until 60 min for the re-equilibration of the system. The column temperature was 30 °C, the flow rate was 0.35 mL/min, and the injection volume was 5 μL. Quantitation of the paracetamol samples was performed using a diode-array detector (Diode RA, UltiMate 3000, Thermo-Scientific, Waltham, MA, USA), operated at a wavelength of 254 nm. Prior to analysis, samples of paracetamol powder and particles were dissolved in a mixture of water–methanol (85:15 *v*/*v*) at a concentration of 0.5 mg/mL, and then filtered through 0.22 μm PTFE filters.

### 3.6. Surface Roughness Measurements

A multi-mode atomic force microscope (AFM, Bruker Multimode AFM, Santa Barbara, CA, USA), operated in a contact mode, was used to obtain a quantitative and qualitative evaluation of the samples. In contact mode, the force between the AFM tip and the sample surface was kept constant by the microscope feedback system while the sample surface was scanned beneath the AFM tip, and the vertical piezoelectric ceramic movement was recorded. Images with 512 × 512 pixels were acquired with a scan size of 50 µm × 50 µm and a scan rate of 2.03 Hz. An NP-type V-shape Si3N4 cantilever (Bruker) with a normal bending constant of k = 0.06 N/m and a tip radius of approximately 0.6 µm was used. During imaging, the set point was chosen to be 2.0 V higher than the top–bottom laser photo detector output obtained when the tip was out of surface contact. AFM images were analyzed using specialized software (Gwyddion 2.54 software). The AFM obtained a 3-dimensional image of the surface of the samples. Eight different areas were measured in each sample at different sections, all located in the center of the samples. The mean roughness (Ra) was recorded after measurement.

### 3.7. Powder X-ray Diffraction (PXRD)

Powder X-ray diffraction analyses were performed using a Stoe StadiP 611 instrument (Stoe, Darmstadt, Germany) equipped with a Mythen1K Si-strip detector. Measurements were performed in transmission geometry with Cu-Kα1 radiation source generated at 40 kV and 40 mA. Samples were scanned with an angular resolution of 0.03° 2θ over a 2θ range from −36° 2θ to +36° 2θ with measurement times of 30 s/PSD-step and a PSD step width of 0.09° 2θ. After the measurements, the diffractograms were folded to range from 0° 2θ to 36° 2θ.

### 3.8. High-Speed Images

A digital high-speed camera (Phantom v12, Phantom High-Speed Cameras—Vision Research, Wayne, NJ, USA) equipped with a microscopic lens (Nikon 10 × CFI plan achromat) was used to observe and document the ejection, flight, impact, and stabilization of the droplet in detail, shooting at 27,000–43,000 fps.

### 3.9. Particle Stability Analysis

Printed paracetamol particles were immediately stored under various conditions and evaluated after 4 weeks to understand further whether the melt-based process affects the API’s stability. Storage conditions of 2–8 °C, 25 °C/62% RH, 40 °C/0% RH, 40 °C/75% RH followed the guidelines of the European Medicine Agency (ICH Q1A (R2) Stability testing of new drug substances and drug products—Scientific guideline) [40]. Storage at 2–8 °C was achieved by storing the particles in an airtight scintillation vial inside a refrigerator. Then, 40/“0” was achieved using a vacuum oven set at 40 °C. Saturated salts were used for the 25/60 and 40/75 conditions. DSC, HPLC, XRD, and SEM analyses at each time point were performed as described above.

## 4. Conclusions

Melt-jet printing technology enabled the production of spherical paracetamol microparticles without the need for solvents or excipients. The printed particles maintained the polymorphic structure of paracetamol and exhibited minimal signs of degradation. The study provides valuable insights into the influence of the printing substrate properties on melt printing outcomes, highlighting the critical role of surface characteristics in achieving the desired API microparticle shapes. Notably, the aluminum substrate yielded near-perfect spherical particles. These findings may contribute to the development and optimization of effective melt printing techniques, especially for small-scale pharmaceutical applications. Using a single printhead, one can produce 1–1.5 mg of spherical particles per minute, multipliable as a function of the number of printing heads in the system.

## Figures and Tables

**Figure 1 pharmaceutics-15-02026-f001:**
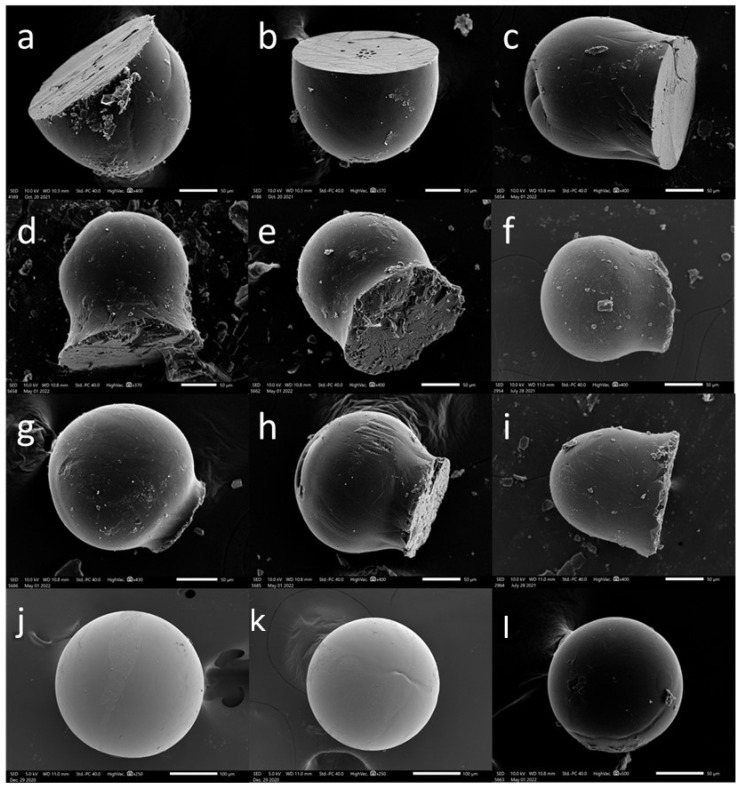
Melt-jet printing of paracetamol on different surfaces. SEM images of microparticles printed on (**a**–**c**) PTFE, (**d**–**f**) glass, (**g**–**i**) PE, and (**j**–**l**) aluminum surfaces; their measured apparent contact angles (sphericity) are detailed in the text. Scale bars: 50 μm.

**Figure 2 pharmaceutics-15-02026-f002:**
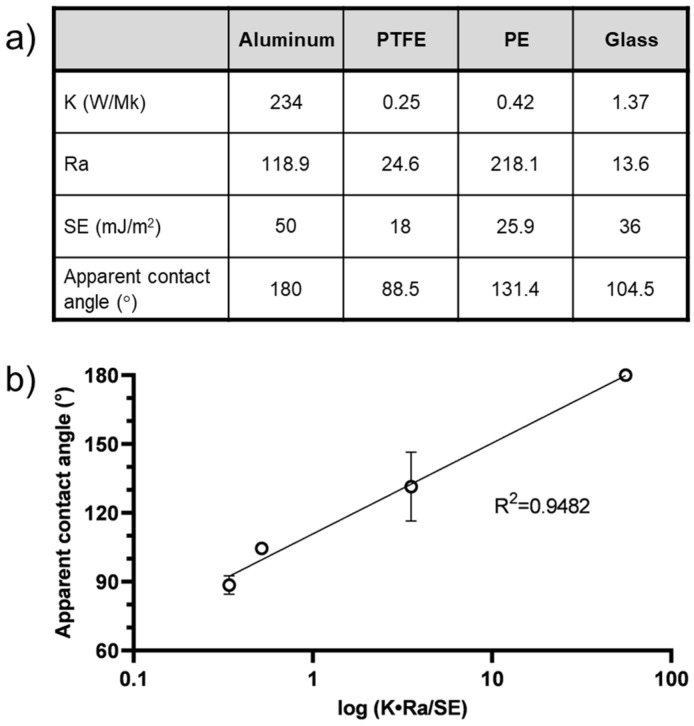
Surface properties and sphericity of paracetamol particles. (**a**) Thermal conductivity (K), roughness (Ra), and surface energy (SE) of aluminum, PTFE, PE, and glass surfaces and corresponding average apparent contact angle of printed paracetamol particles. (**b**) Relationship between the sphericity of paracetamol particles (apparent contact angle) and an expression incorporating surface properties (K, Ra, and SE).

**Figure 3 pharmaceutics-15-02026-f003:**
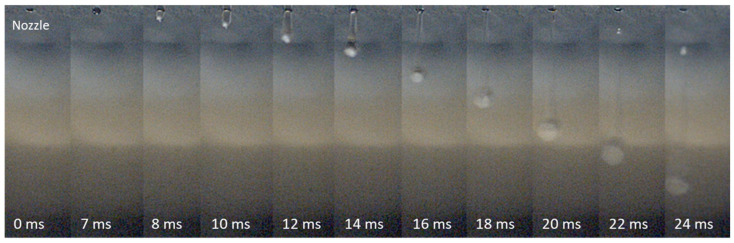
Melt-jet printing process paradigm: a sequence of images (**from left to right**) captured with a high-speed digital camera, showing the ejection of a molten paracetamol droplet from the nozzle (**top**) on an aluminum surface (**bottom**).

**Figure 4 pharmaceutics-15-02026-f004:**
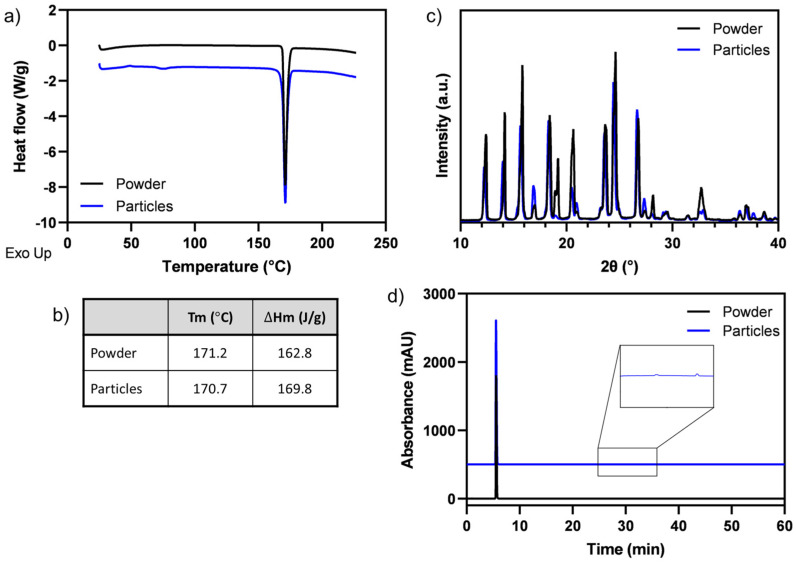
Paracetamol microparticles characterization. (**a**) DSC thermograms, (**b**) melting temperatures (Tm) and enthalpies (∆Hm), (**c**) XRD patterns, and (**d**) HPLC chromatograms of neat powder (black) and printed particles (blue).

**Figure 5 pharmaceutics-15-02026-f005:**
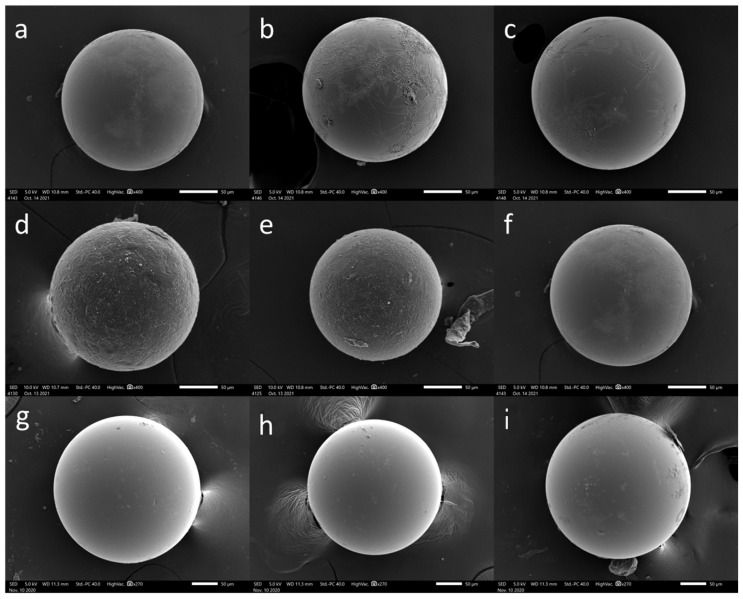
API particles printed on an aluminum surface. SEM images of (**a**–**c**) paracetamol, (**d**–**f**) carbamazepine, and (**g**–**i**) indomethacin microparticles. Scale bars: 50 μm.

## Data Availability

The data presented in this study are available in this article.

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
