# Peer review of "Controlling Microparticle Morphology in Melt-Jet Printing of Active Pharmaceutical Ingredients through Surface Phenomena"

_pharmaceutics, 2023, doi:10.3390/pharmaceutics15082026_

Round 1

Reviewer 1 Report

The manuscript “Controlling Microparticle Morphology in Melt-Jet Printing of Active Pharmaceutical Ingredients Through Surface Phenomena” describes the next study of the authors on the melt-jetting of pharmaceutical materials, in this case, pure API. It aims to produce spherical particles. Production of spherical particles, especially API particles, is an important issue regarding pharmaceutical technology.

The manuscript requires some clarifications and some minor improvements before publication.

„Melt-Jet Printing” in the title of the manuscript is somewhat strange. In my opinion, just “melt-jetting” is relevant. I’m aware that adding the “printing” term makes the method (and title) more attractive, but I think it is misleading. What pattern/structure was printed? I cannot find any. Melt-jetting, in this case, is the method for obtaining spherical particles ( “... allows for the fabrication of solid spherical particles from melts.”).

What is the real goal of the study? Preparation of ingredients for further processing aiming preparation of final dosage forms? Or the produced spherical particles are dosage forms themselves? It should be clarified.

What is the yield of the process? The authors did not touch on the issue.

Figure 3 is not clear – what are the regions of different grayscale? Where is the nozzle? Where is the substrate?  

Author Response

1) Melt-Jet Printing” in the title of the manuscript is somewhat strange. In my opinion, just “melt-jetting” is relevant. I’m aware that adding the “printing” term makes the method (and title) more attractive, but I think it is misleading. What pattern/structure was printed? I cannot find any. Melt-jetting, in this case, is the method for obtaining spherical particles ( “... allows for the fabrication of solid spherical particles from melts.”).

We agree that Melt-Jetting covers most of the technology’s paradigm, yet the one difference we think should not be overlooked is the critical role of the surface itself in the process. Hence the deposition of liquid droplets onto a surface resembles a “classic” inkjet printing process thus, we named the process “melt-jet printing” and not just “melt jetting”.

2) What is the real goal of the study? Preparation of ingredients for further processing aiming preparation of final dosage forms? Or the produced spherical particles are dosage forms themselves? It should be clarified.

The goal of the study was to produce API particles with controlled properties for use in the preparation of dosage forms, as clarified in the Introduction section (page 3, lines 98-102).

3) What is the yield of the process? The authors did not touch on the issue.

The printing head is set to 30-50 Hz. Each droplet weighs about five µg. Thus, we produce ≈1.2 mg/min. Since the particles do not adhere to the surface, they are easily removed; therefore, we did not encounter any limitations in the printable area (we use 20 cm diameter aluminum weighing dishes). We've added to the text an additional comment regarding throughput (page 11, lines 377-378).

4) Figure 3 is not clear – what are the regions of different grayscale? Where is the nozzle? Where is the substrate?  

The figure was edited and clarified accordingly in the manuscript.

Reviewer 2 Report

Bornstein and co-workers have done a great job in Investigating "Controlling Microparticle Morphology in Melt-Jet Printing of Active Pharmaceutical Ingredients Through Surface Phenomena." Reviewer has very minor comments to address, as following:

1) Please comment on the applicability of this technology to thermally sensitive API's? 

2) How much time does it take to print certain quantity of API? For e.g., how long it will take to print each (1) gram of API? and How much surface area it takes to print per each (1) gram of API? 

3) Understand that this is still in initial stage of development, but can this be scalable? 

4) Please provide all the stability data associated with printed API, for e.g., long term stability.

5) Please submit any data on surface morphology of the printed API comparing it to longterm storage (at least 1 month), to demonstrate that the printed API still retain the desired surface morphology.

Author Response

1) Please comment on the applicability of this technology to thermally sensitive API's? 

The thermal exposure above the melting point could affect the structure and integrity of thermally sensitive APIs; a comment indicating that was added to the results and discussion section (Page 7, Lines 253-254). 

2) How much time does it take to print certain quantity of API? For e.g., how long it will take to print each (1) gram of API? and How much surface area it takes to print per each (1) gram of API? 

The printing head is set to 30-50 Hz. Each droplet weighs about five µg. Thus, we produce ≈1.2 mg/min. Since the particles do not adhere to the surface, they are easily removed; therefore, we did not encounter any limitations in the printable area (we use 20 cm diameter aluminum weighing dishes). We've added to the text an additional comment regarding throughput in the conclusions (page 11, lines 377-378).

3) Understand that this is still in initial stage of development, but can this be scalable? 

Scalability can be achieved by the multiplication of printing heads or by the multiplication of orifices from a single printing apparatus. This was addressed in the conclusions in line with comment number 2.

4) Please provide all the stability data associated with printed API, for e.g., long term stability.

5) Please submit any data on surface morphology of the printed API comparing it to longterm storage (at least 1 month), to demonstrate that the printed API still retain the desired surface morphology.

We performed storage analyses at various conditions, including two from the European Medicine Agency guidelines. They were originally not added because of our notion that these have a minor contribution to the article's coherency. Per the reviewer's request, four weeks of storage data (HPLC, DSC, XRD, SEM) under various conditions have been added to the manuscript (page 7 lines 255-262, page 11 lines 358-367) and to the supplementary material (Figure S2).

Round 2

Reviewer 2 Report

Thanks and no further comments